# Automatic Evaluation vs. User Preference in Neural Textual Question Answering over COVID-19 Scientific Literature

**Arantxa Otegi, Jon Ander Campos, Gorka Azkune,**
**Aitor Soroa, Eneko Agirre**
University of the Basque Country (UPV/EHU)
{arantza.otegi, jonander.campos, gorka.azkune,
a.soroa, e.agirre}@ehu.eus

## Abstract

We present a Question Answering (QA) system that won one of the tasks of the Kaggle CORD-19 Challenge, according to the qualitative evaluation of experts. The system is a combination of an Information Retrieval module and a reading comprehension module that finds the answers in the retrieved passages. In this paper we present a quantitative and qualitative analysis of the system. The quantitative evaluation using manually annotated datasets contradicted some of our design choices, e.g. the fact that using QuAC for fine-tuning provided better answers over just using SQuAD. We analyzed this mismatch with an additional A/B test which showed that the system using QuAC was indeed preferred by users, confirming our intuition. Our analysis puts in question the suitability of automatic metrics and its correlation to user preferences. We also show that automatic metrics are highly dependent on the characteristics of the gold standard, such as the average length of the answers.

## 1 Introduction

The global health crisis caused by COVID-19 raised several challenges not only to medical research community but also to the Natural Language Processing (NLP) community. The pace of publication of scientific articles on this topic has caused an information overload for health experts, highlighting the need for information systems based on NLP and artificial intelligence that assist in finding relevant and accurate information in publications.

Faced with this situation, the White House and several leading research groups released the COVID-19 Open Research Dataset (CORD-19) (Wang et al., 2020), which contains over 200K articles about COVID-19, SARS-CoV-2 and related coronaviruses. Additionally, they launched the Kaggle CORD-19 Challenge where they defined 10 tasks with the aim of developing text and data mining tools on top of this dataset that can help the medical community to find answers to high priority scientific questions.

The challenge has been a real success with more than 500 teams participating in the first round of the challenge (Wang et al., 2020). In this paper we present a Question Answering (QA) system that won one of the tasks in Round One of the Kaggle CORD-19 Challenge. Our goal when building the system was to fight information overload. For this purpose, the system highlights sub-sentence replies, and only answers when the quality of retrieval and answers is satisfactory. The system uses neural textual QA techniques to directly find specific answers to the scientific questions listed in the tasks of the challenge. The system is not tailored towards specific questions, and can be readily used to answer any other question.

One of the challenges during development was the lack of resources for automatically evaluating the system. After submission we have been able to partially evaluate the main components of our system, using related shared-tasks and datasets that have been recently released and were not available at the time. The automatic evaluation revealed that the submitted system was not optimal, contradicting some design choices, such as using the QuAC dataset (Choi et al., 2018) to fine-tune the QA system in addition to the SQuAD 2.0 (Rajpurkar et al., 2018) dataset. We thus performed an additional A/B test to check whether users preferred the system using QuAC or not. The test confirmed our intuition, and raises questions on the suitability of automatic metrics and its correlation to human satisfaction.

Our findings are in line with de Vries et al. (2020), who claim that due to the tradition of prioritizing quantity of data to naturalness and real-world use cases, researchers could end up with findings that are not relevant for the real world. In this

sense the system trained on the SQuAD 2.0 dataset, which is not built with prospective users, achieves highest automatic metrics but falls behind when attending to user preference. There have been recent efforts for building QA datasets closer to real-world use cases (Choi et al., 2018; Reddy et al., 2019; Campos et al., 2020; Castelli et al., 2020). Among all of them, we show that the system trained also on the QuAC dataset is preferred by the users.

The contributions of our paper are two-fold. First, we describe the QA system that is able to answer COVID related questions to aid clinical scientists to find the information they need effectively among thousands of scientific papers. Second, we report some analysis on the shortcomings that automatic evaluation of QA systems suffer from.

## 2 Related Work

The release of the CORD-19 dataset has attracted the interest of a wide variety of researchers, who have focused on developing systems that help other scientists to obtain relevant information about the COVID-19.

Some of these systems, as the one developed by *Verizon Media* [1], focus themselves on the retrieval step. In this case they propose a search engine over the whole CORD-19 dataset using the Vespa data serving engine. More similar to ours we have the CORD-19 search [2] developed with Amazon Comprehend Medical and the COVID19 Research Explorer [3] developed by Google. In both of these systems, apart from retrieving articles with the search engine, they also highlight the most probable answer in the retrieved paper.

Development of all these systems is contemporary to the development of ours so we all share similar ideas but with different design patterns. Apart from that, all the previously mentioned systems lack any quantitative evaluation.

## 3 System Description

In this section we describe each of the modules of our neural QA system which answers COVID-19 related queries. The system is composed of two main sub-systems. The first is an Information Retrieval (IR) module, which, given a user question, searches among the whole dataset and selects a

---

[1] https://cord19.vespa.ai/
[2] https://cord19.aws
[3] https://covid19-research-explorer.appspot.com/

short list of candidate documents that may contain the answer. The second is a QA module that searches for answers to the original question on the documents obtained in the first step.

**Datasets** The system relies on the freely available CORD-19 dataset of scientific papers about COVID-19, SARS-CoV-2 and related coronaviruses (Wang et al., 2020) to extract the answers. Two datasets are used to fine-tune the neural QA module: SQuAD2.0 (Rajpurkar et al., 2018), which is a reading comprehension dataset widely used in the QA research community, and QuAC (Choi et al., 2018), which is a Conversational QA dataset containing a higher rate of non-factoid questions than SQuAD.

**Paper Filtering** As we are mostly interested in papers related to COVID-19, we filter out papers that are about coronaviruses other than COVID-19 (for example, SARS-CoV and MERS) from the CORD-19 dataset. For that purpose, we created a list of synonyms of COVID-19 ("coronavirus 2019", "ncov 2019", "sarscov2", "wuhan coronavirus", etc), and we check if a synonym appears in the title or the abstract of a paper. In that way, we filter out those papers that do not include any of the synonyms. From now on, we will consider only the papers that we keep after filtering.

**Information Retrieval** The IR module indexes not only the abstracts, but also the full text of the papers. As shorter piece of texts are more favorable for the following QA component in the pipeline, the indexing unit is an abstract or each of the paragraphs of the full text. The text is tokenized, stemmed and lower cased, and a stopword filter is applied before indexing as common practice. The classical BM25F search algorithm (Zaragoza et al., 2004) is used to retrieve the most relevant paragraphs given a natural language question.

**Question Answering** Given a question in natural language and a paragraph, the QA module returns the answer to the question in the paragraph or "No answer" otherwise. The implemented system is based on neural network techniques. More specifically, we have used the SciBERT language representation model, which is a pretrained language model based on BERT, but trained on a large corpus of scientific text, including text from biomedical domain (Beltagy et al., 2019). Following the usual reading comprehension method we use SciBERT

as a pointer network, which selects an answer start and end index given a question and a paragraph. We used both SQuAD and QuAC to fine-tune SciBERT for QA. Manual inspection revealed that a system fine-tuned on SQuAD2.0 produced answers that were specially good for COVID related questions seeking short answers. However, we also observed that a fine-tuning with SQuAD2.0 and QuAC produced answers of better quality, particularly for questions which require longer answers. We thus decided to use a SciBERT model fine-tuned first on SQuAD2.0 and then on QuAC[4] as our final model.

**Pipeline** The final system [5] combines both IR and QA modules in a pipeline, as follows. First, we refined (split or simplify) manually the questions that were too large and complex for the QA system. Then we retrieved the 20 most relevant paragraphs using the IR module for each of the questions. Finally, we ran the QA module over the relevant paragraphs to select specific answers from them. If more than %85 of these 20 answers are of "No answer" type, the current question will not be answered to not overload the users with bad answers. In the other cases, the best answer for each of the best five paragraphs will be shown. Additionally, next to each answer, we show extra information such as the journal and title of the paper (with a link to access online version on the web). Moreover, we also show the paragraph from which each answer has been extracted. Each paragraph is highlighted with the best 3 answers, using different lightness of color (the darker the better the answer). Appendix A shows an example of the output of the system for one of the questions of a task from Kaggle CORD-19 Challenge.

## 4 Discussion on Evaluation Methods

The systems in the Kaggle challenge were judged by an evaluation committee based on the accuracy, as well as on the documentation and the overall presentation of the submissions. No automatic evaluation was performed, due to the lack of manually annotated datasets available. According to this manual evaluation, our system was the best on one of the tasks of the CORD-19 Challenge.

After the challenge finished several related datasets have been released, and therefore we have

---

[4] https://www.kaggle.com/jonander95/bertsquadquac

[5] https://www.kaggle.com/aotegi/neural-question-answering-for-cord19-task8

| Run | P@5 | NDCG@10 | MAP | **bpref** |
|---|---|---|---|---|
| sab20.1.meta.docs | 78.0 | 60.8 | 31.3 | 48.3 |
| sab20.1.merged | 62.7 | 51.1 | 24.1 | 48.2 |
| UIowaS_Run3 | 64.7 | 52.9 | 26.2 | 46.9 |
| ... | ... | ... | ... | ... |
| ixa-ir-filter-query | 56.7 | 44.0 | 19.7 | 40.0 |

Table 1: Results of the top 3 ranked systems in round 1 of the TREC-COVID Challenge ranked according to bpref metric. Last row shows the performance of our IR system.

been able to perform an automatic quantitative evaluation of the system. In this section we present the results of the automatic evaluation of both IR and QA modules, as well as an additional human evaluation carried out in-house with the purpose of assessing the correlation of automatic evaluation metrics with human satisfaction.

### 4.1 Evaluation of the IR Module

We evaluated the IR module by submitting it to the TREC-COVID Challenge (Roberts et al., 2020). This challenge is an IR shared-task for COVID-19 based on the CORD-19 dataset. The challenge is divided in 5 rounds, but we submitted our system only to the first one, where systems were evaluated based on 30 COVID related topics. Our system ranked 28th among the 100 automatic systems when using binary preference-based (bpref) metric (Buckley and Voorhees, 2004) for evaluation. Table 1 shows some of the results of the automatic systems. [6]

### 4.2 Evaluation of the QA Module

**Automatic evaluation** For the automatic evaluation of the different QA models we are using the COVID-QA dataset (Möller et al., 2020) that is one of the few publicly available QA datasets related with the disease. This dataset has been annotated by 15 biomedical experts taking 147 articles from the CORD-19 dataset and has a total of 2,019 question/answer pairs. We use this dataset for zero-shot evaluation with different transformer based language models fine-tuned on the SQuAD and QuAC datasets. We used the standard F1 and Exact Match (EM) measures for evaluating the systems.

The results of the transformer based language models can be seen in Table 2. Here we compare the baselines based on RoBERTa proposed in Möller et al. (2020) to our fine-tuned SciBERT

---

[6]For more detailed information you can access https://ir.nist.gov/covidSubmit/index.html

based models. RoBERTa models slightly outperform BERT on average, which is somehow expected and consistent with previous results (Liu et al., 2019). The table shows that adding the QuAC dataset at fine tuning phase causes the results to drop in both EM and F1 by $\sim 10$ and $\sim 20$ points, respectively. This result contradicts previous manual analysis that showed the model tuned on both SQuAD and QuAC producing preferable results, and which guided our decision to using it in the Kaggle challenge. We performed an additional manual analysis to shed light on this apparent contradiction, which we described in the following section.

**Manual analysis** We performed an in-house test for comparing the user preference among the answers given by the two different SciBERT systems that have been fine-tuned with SQuAD and SQuAD + QuAC. We designed an A/B test where two annotators were provided with the correct (gold) answer together with the top 1 ranked answers from each model. The task consisted on selecting the most suitable answer between the two options, as well as "none" if both answers were equally correct or incorrect. After 50 annotations, the model fine-tuned on SQuAD + QuAC was preferred 18.5 times on average, the model fine-tuned on SQuAD only once, and the rest were ties. Both annotators had an agreement rate of 86%. This confirms our initial analysis, and shows a mismatch between human preference and automatic metrics.

In order to better understand this phenomena, we focused on the cases where the SQuAD + QuAC model was selected, as shown in Figure 1. Here, we can spot a trend in which the annotator prefers longer self-explanatory answers than short facts. We posit that these longer answers can help the users in order to trust the system, as short facts are many times impossible to contrast and difficult to trust. However, automatic metrics such as F1 heavily penalize long answers, as they overlap poorly with the gold annotations, which are mostly short, factual answers. Figure 2 shows the answer length distribution of the gold standard and the two models. The model fine-tuned in SQuAD produces answers whose length is much more similar to the gold standard which explains the results of Table 2.

This analysis reveals that automatic metrics are highly dependent on the particular guidelines followed when annotating the datasets. If annotators are asked to select short answers, systems that pro-

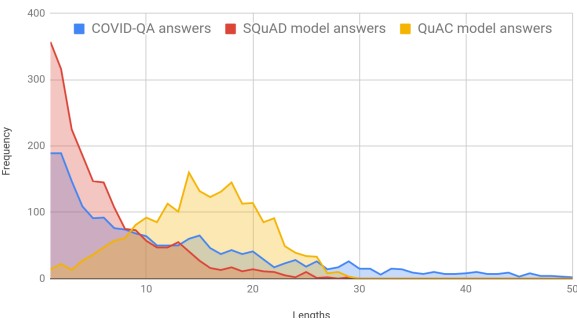

| Examples | F1 |
|---|---|
| **Question**: *How many known species of Rotavirus exist?* | |
| **Gold**: *nine species* | - |
| **SQuAD**: *nine* | 0.66 |
| **SQuAD + QuAC**: *The rotavirus belongs to the Reoviridae family and comprises nine species known as Rotavirus group* | 0.22 |
| **Question**: *When did the first known cases of Middle East respiratory syndrome (MERS) occur?* | |
| **Gold**: *in 2012* | - |
| **SQuAD**: *2012* | 0.66 |
| **SQuAD + QuAC**: *occurred in 2012 in Jordan but were reported retrospectively.* | 0.36 |

Figure 1: Examples where the SQuAD + QuAC system answer was preferred by the human but penalized hardly by the F1 metric.

Figure 2: Distribution of answers length for the COVID-QA dataset and the two compared systems.

duce longer answers are severely penalized, even if these answers are satisfactory for the users.

## 5 Conclusions and Future Work

We have presented a QA system that is able to answer COVID-19 related questions with a high degree of success, as shown by the prize obtained at the Kaggle CORD-19 Challenge. In this challenge a qualitative evaluation was conducted by a committee formed by experts, but no automatic evaluation was performed. We present an automatic evaluation of both the IR and QA modules, using different resources that have been created recently. In the case of the use (or not) of QuAC training data, the automatic evaluation contradicted our intuitions, and we thus performed A/B test between the system submitted to Kaggle (using QuAC) and the system that performed best according to the automatic evaluation (not using QuAC). This A/B test confirms our initial intuitions, and shows that the automatic evaluation metrics do not correlate well with human preference. The analysis also shows that automatic metrics are highly dependent on the characteristics of the gold standard, such as the average length of the answers.

| Model | EM | F1 |
|---|---|---|
| RoBERTa base | 21.84 | 49.43 |
| RoBERTa base + SQuAD | 25.90 | 59.53 |
| SciBERT base + SQuAD | 28.72 | 51.01 |
| SciBERT base + SQuAD + QuAC | 7.52 | 40.87 |

Table 2: Results obtained by different baselines on the COVID-QA dataset doing zero-shot experiments.

As future work, we plan to perform more thorough studies on the contradictions between automatic evaluation and user preferences, in line with de Vries et al. (2020). We would also like to devise automatic metrics for QA in scenarios where complex and self-explanatory answers are expected by the user, hopefully showing higher correlation with user preferences. These metrics would be essential for the research community in order to build tools that are helpful for end users.

## Acknowledgments

This research was partially supported by VIGI-COVID project FSuperaCovid-5 (Fondo Supera COVID-19 / CRUE-CSIC-Santander), project DeepReading (RTI2018-096846-BC21) supported by the Ministry of Science, Innovation and Universities of the Spanish Government, project DeepText (KK-2020/00088), the Basque Government excellence research group (IT1343-19) and the NVIDIA GPU grant program. Jon Ander Campos enjoys a doctoral grant from the Spanish MECD.

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

## A   System Output Example

### WHAT DO WE KNOW ABOUT DIAGNOSTICS AND SURVEILLANCE?

**Is the use of screening of neutralizing antibodies such as ELISAs valid for early detection of disease?**

**In a study of 623 sars patients , the neutralizing - antibody levels peaked at 20 - 30 days and were sustained for over 150 days .** [Pathogenesis of severe acute respiratory syndrome, *Current Opinion in Immunology*, 2005-08-31]

Detection of serum IgG , IgM and IgA against SARS - CoV using immunofluorescent assays and by ELISA against nucleocapsid antigen occurs around the same time with most patients seroconverted by day 14 after onset of illness [ 48 ] . IgG can be detected as early as 4 days after the onset of illness . The kinetics of neutralization antibodies nearly parallel those for IgG [ 48 ] and most of the neutralizing - antibody activity is attributed to IgG [ 49 ] . In a study of 623 SARS patients , the neutralizing - antibody levels peaked at 20 - 30 days and were sustained for over 150 days . These antibodies can neutralize the pseudotype particles bearing the S protein from different SARS - CoV strains , suggesting that these antibodies are broadly active and that the S protein is highly immunogenic [ 49 ] . Indeed the S protein , among the other structural proteins , such as M , E or N , is the only significant SARS - CoV neutralization antigen and protective antigen [ 50 ] , with amino acids 441 - 700 as the major immunodominant epitope [ 51 ] .

**Early antibodies are detected in some patients within two weeks .** [Severe acute respiratory syndrome and dentistry A retrospective view, *The Journal of the American Dental Association*, 2004-09-30]

Enzyme - linked immunosorbent assay , or ELISA , test . From about 20 days after the onset of clinical signs , ELISA tests can be used to detect immunoglobulin , or Ig , M and IgA antibodies in the serum samples of patients with SARS . Early antibodies are detected in some patients within two weeks .

**Serologic assays are not useful for early diagnosis as igg antibodies do not appear for 7 - 10 days after onset of symptoms .** [SARS: future research and vaccine, *Paediatric Respiratory Reviews*, 2004-12-31]

Serologic assays are not useful for early diagnosis as IgG antibodies do not appear for 7 - 10 days after onset of symptoms . It has been stated that IgM antibodies typically appear earlier , but detection of IgM antibodies does not appear to permit earlier diagnosis . 1 , 11 Since a few SARS patients have had late seroconversion , it is best to test the convalescent serum collected at least 21 days and preferably 28 days after onset of symptoms , to rule out SARS . 1 At present , the most widely used methods for detection of antibodies against SARS CoV are indirect immunofluorescence assay and ELISA with cell - culture extract , which are difficult to standardise . 25 Therefore , recombinant - antigenbased ELISA assays are being developed using highly immunogenic nucleocapsid protein of SARS CoV , which can be used for a large scale epidemiological study of seroprevalence . 25

Figure 3: System responses given to one of the questions on the topic "What do we know about diagnostics and surveillance?". 3 different answers (in bold) are shown for the question, and for each question additional information is given: the title with the link of the relevant paper, the date of the publication and the relevant paragraph where the best answers are highlighted (in orange).

In Figure 3 an example of the system output can be appreciated, as shown to users by the interface. The interface shows the best three answers in context, using highlighting. Instead of highlighting just the tokens of the highest scoring answer span, we use shades of orange to highlight alternative, lower scoring answers in the same passage (if present).