# OpenReview forum: "Automatic Evaluation vs. User Preference in Neural Textual QuestionAnswering over COVID-19 Scientific Literature"
_EMNLP/2020/Workshop/NLP-COVID — NLP-COVID19-EMNLP Oral_

### Official Review · AnonReviewer3 · 2020-09-22
**A good QA system and very well-written paper, but questionable evaluation analysis.**

**Rating:** 6
**Confidence:** 3

**Review:**

The paper presents a question answering system that won one of the tasks in the round one of the Kaggle CORD-19 Challenge. The authors describe the two system components – IR (retrieving the most relevant paragraph given a question in natural language) and QA (extracting the answer from the paragraph). The paper further analyses a surprisingly low performance introduced by the system when fine-tuned on SQuAD and QuAC on an annotated dataset, revealing that the inherent preference of short answers by evaluation systems causes the inspected drop in accuracy.

The paper is interesting, insightful, mostly clear and very well-written.

The main contribution of this work is summarized in section 4.2 – the analysis of automatic evaluation of QA systems w.r.t. short vs. long answers, reflecting essentially the same information. My main concern is that the conclusions are drawn based on 50 or so annotations done by a single annotator, who may well have preferences re short vs. long answers for what essentially are factoid questions. That is, the “user preferences” as stated in the title are somewhat misleading. Which leads me to my second concern – the two examples in figure 1 (“how many” and “when” questions) call for a short fact-based answer IMO, or, at least should rank a short and precise fact providing an exact answer to a query not inferior to a long paragraph including the answer plus some bits of potentially unnecessary information.

Can the entire drop in EM and F1 in the last row in table 2 be attributed to the analysis in section 4.2? Are there any other potential factors that could affect the QA accuracy?

I appreciate the authors’ analysis in figure 2, interesting!

All in all, I think that’s a good work. The fact that the proposed system (when fine-tuned on SquAD only, third row in table 2) outperforms ROBERTA-based solutions highlights the contribution of this work, which is sufficient to recommend for acceptance IMO, despite the two issues above.

---

> ### Author Response · Authors · 2020-09-27
> **Re: A good QA system and very well-written paper, but questionable evaluation analysis.**
>
> Thanks for the insightful feedback.
>
> We do agree with the reviewer when claiming that a further analysis on the EM and F1 drop would be interesting. At this point we spotted that answer lengths could be one major reason for that as shown by the 50 annotations and the answers length distribution. We plan to check other potential factors in the future.
>
> Apart from that, we understand the concern of the reviewer in terms of factoid questions, but all the preliminary human analysis that we have carried out show that even if we have a factoid question the users prefer having more self explanatory answers. We posit that these longer answers can help the users in order to trust the system, as short facts are many times impossible to contrast and difficult to trust. We also run the test through another annotator with an agreement rate of 86%. We will report this in the final version.
>
> We also consider the fact that a system which had won one of the tasks of the highly competitive Kaggle CORD-19 challenge was performing so poorly in terms of F1. We take that as an independent indication of the possible mismatch of automatic measures and user preferences.
>
> So the main aim of this work is to arouse discussion on what users prefer and not in the correctness or incorrectness of the answers. Taking the examples in Figure 1 we do not claim that short fact-based answers are incorrect, but that longer and more self-explanatory answers were preferred by the in-house annotator and the Kaggle evaluators. We will try to clarify this better in the final version.

---

### Official Review · AnonReviewer1 · 2020-09-22
**Interesting paper on the suitability of automatic metrics in the context of QA systems**

**Rating:** 7
**Confidence:** 3

**Review:**

In this work the authors introduce their QA system that won one of the tasks in the round one of the Kaggle CORD-19 Challenge, and also make the case of the suitability of automatic metrics. With an ambitious scope, the authors very effectively frame the problem and their QA system in contrast to others in the same competition. The description of their system is quite clear and succinct, making this a very easy and intuitive read. Section 4.1. seems a loose and needs more refinement, as for example Table 1 just cherry-picks some other systems IR module to compare against (the top 5? what number was each? it just picks 3 out of 50 randomly). The evaluation of the QA system in section 4.2 is quite interesting and while brief, it shows the results are unintuitive and lead to the interesting idea of the suitability of automatic metrics after they performed a manual review. This manual review is a good idea but it is lacking in depth and detail to be fully stand-alone, more discussion would be nice to have. Overall, the paper proposes an interesting system and poses a very interesting question with regards the typical evaluation procedures for such systems, while it needs a bit of work and tidying up, it is a nice contribution.

---

> ### Author Response · Authors · 2020-09-27
> **Re: Interesting paper on the suitability of automatic metrics in the context of QA systems**
>
> Thanks for the insightful review.
>
> Attending section 4.1 we just reported 3 randomly picked systems out of 100 in order to have an overview of the performance range of different runs without taking a lot of space in the manuscript. We then suggest the readers to access the official ranking in the link of the footnote. However, we understand your concerns and we will select just the top 3 systems together with ours to appear on the paper.
>
> For section 4.2 we do agree that further discussion would be really interesting as the correct automatic evaluation of QA systems is a main concern of the community. Our idea with this paper was to show an example of a real case where user preference seems to mismatch automatic metrics and with that we wanted to arouse further discussion. We will try to clarify this better in the final version.

---

### Official Review · AnonReviewer2 · 2020-09-24
**A well-written paper for a QA system for COVID-19 question answering**

**Rating:** 7
**Confidence:** 3

**Review:**

The paper describes a COVID-19 QA system. It uses BM25 for the IR component and BERT models for the QA component. It provides both quantitative evaluations on the COVID-QA dataset and in-depth manual error analysis. The manuscript is well-written and the structure is clear. Two comments are summarized below.

1. It would be more robust to provide the performance of baseline systems and other systems on the COVID-QA dataset. This will make it clear on whether the proposed system is indeed effective.

2. The manual analysis part is interesting. More details are expected: any suggested changes on the evaluation metrics? What are the recommendations from the study? This is critical for QA systems in production.

---

> ### Author Response · Authors · 2020-09-27
> **Re: A well-written paper for a QA system for COVID-19 question answering**
>
> Thanks for the insightful feedback.
>
> 1. The RoBERTa model is the baseline proposed in the COVID-QA dataset. We are not aware of more results reported on this dataset but we will add them to the manuscript if we find any.
>
> 2. These are great suggestions, we are working on that and hope to have specific recommendations soon.